# Evidence for Epistatic Interaction between *HLA-G* and *LILRB1* in the Pathogenesis of Nonsegmental Vitiligo

**DOI:** 10.3390/cells12040630

**Published:** 2023-02-15

**Authors:** Maria Luiza de Oliveira-Caramez, Luciana Veiga-Castelli, Andreia S. Souza, Renata Nahas Cardili, David Courtin, Milena Flória-Santos, Eduardo Donadi, Silvana Giuliatti, Audrey Sabbagh, Erick C. Castelli, Celso Teixeira Mendes-Junior

**Affiliations:** 1Departamento de Genética, Faculdade de Medicina de Ribeirão Preto, Universidade de São Paulo, Ribeirão Preto 14049-900, SP, Brazil; 2Molecular Genetics and Bioinformatics Laboratory, School of Medicine, São Paulo State University (UNESP), Botucatu 18618-687, SP, Brazil; 3Departamento de Clínica Médica, Faculdade de Medicina de Ribeirão Preto, Universidade de São Paulo, Ribeirão Preto 14049-900, SP, Brazil; 4Institut de Recherche pour le Développement, UMR 261 MERIT, Université de Paris, F-75006 Paris, France; 5Departamento de Enfermagem Materno-Infantil e Saúde Pública—(EERP/ERM), Universidade de São Paulo, Ribeirão Preto 14049-900, SP, Brazil; 6Pathology Department, School of Medicine, São Paulo State University (UNESP), Botucatu 18618-687, SP, Brazil; 7Laboratório de Pesquisas Forenses e Genômicas, Departamento de Química, Faculdade de Filosofia, Ciências e Letras de Ribeirão Preto, Universidade de São Paulo, Ribeirão Preto 14040-901, SP, Brazil

**Keywords:** ancestry, Brazil, ILT-2, ILT-4, LILRB2, SNP*for*ID

## Abstract

Vitiligo is the most frequent cause of depigmentation worldwide. Genetic association studies have discovered about 50 loci associated with disease, many with immunological functions. Among them is HLA-G, which modulates immunity by interacting with specific inhibitory receptors, mainly LILRB1 and LILRB2. Here we investigated the *LILRB1* and *LILRB2* association with vitiligo risk and evaluated the possible role of interactions between HLA-G and its receptors in this pathogenesis. We tested the association of the polymorphisms of *HLA-G*, *LILRB1*, and *LILRB2* with vitiligo using logistic regression along with adjustment by ancestry. Further, methods based on the multifactor dimensionality reduction (MDR) approach (MDR v.3.0.2, GMDR v.0.9, and MB-MDR) were used to detect potential epistatic interactions between polymorphisms from the three genes. An interaction involving rs9380142 and rs2114511 polymorphisms was identified by all methods used. The polymorphism rs9380142 is an *HLA-G* 3′UTR variant (+3187) with a well-established role in mRNA stability. The polymorphism rs2114511 is located in the exonic region of *LILRB1*. Although no association involving this SNP has been reported, ChIP-Seq experiments have identified this position as an EBF1 binding site. These results highlight the role of an epistatic interaction between *HLA-G* and *LILRB1* in vitiligo pathogenesis.

## 1. Introduction

Vitiligo is characterized by white spots on the skin, which arise due to the dysfunction of melanocytes [1]. Based on clinical presentation, the disease can be classified as segmental vitiligo, with unilateral distribution, affecting 10–15% of the cases, and shows an early age of onset; or nonsegmental vitiligo, with bilateral and symmetrical distribution, is the most common manifestation, representing almost 80% of the cases, besides several other forms that may fit into the spectrum of rare vitiligo [2,3]. It is considered the most frequent cause of depigmentation worldwide, without preference for ancestry or sex. It presents marked geographic differences in prevalence, which is mostly about 0.5% to 2% but can be up to 4.7% in Nigeria and more than 8% in India. Meanwhile, the reasons underlying this heterogeneous distribution still need to be clarified [4,5].

Studies have evidenced a polygenic/multifactorial mode of inheritance for nonsegmental vitiligo, with about 50 loci already identified, many of them with roles in immunological response and previously associated with other autoimmune diseases [6]. A pivotal role of the immune system in nonsegmental vitiligo is sustained by the presence of autoantibodies against melanin and the higher frequency of other autoimmune conditions in vitiligo patients [7].

Notwithstanding, only 50% of the total heritability has been identified in nonsegmental vitiligo [8,9]. A substantial amount of this variance thus remains unexplained, commonly referred to as the “missing heritability” [10]. It has been suggested that missing heritability may be partially due to gene–gene interactions, also called epistasis [11,12,13]. Epistasis occurs when two or more loci influence a phenotype in a dependent manner and may partly explain why human mapping can be difficult to replicate [14]. Additionally, it is expected (and observed) that genes that exhibit physical interactions also demonstrate these interactions at the genetic level [15].

The binding of HLA-G to its receptors, LILRB1 and LILRB2, induces the downregulation of innate and adaptive immunity, thus modulating the activity of NK cells, antigen-presenting cells, and T and B lymphocytes [16,17,18].

Based on the critical role of HLA-G in the immune response and on the associations previously reported between *HLA-G* polymorphisms and diverse autoimmune conditions, our group previously evaluated associations with the whole *HLA-G* locus with vitiligo in the Brazilian population, being observed for the first time an association between the rs9380142 SNP (+3187G) located in 3′UTR (untranslated region) and disease risk [19]. Nonetheless, up to this moment, no study has investigated the associations of *LILRB1/LILRB2* variants with the disease yet.

Since genes encoding interacting proteins are likely to evolve to preserve proper protein interactions, as well as an appropriate stoichiometry among its interacting components [20], we employed the multifactor dimensionality reduction (MDR) method to evaluate the possible role of epistatic interactions between HLA-G and its receptors in the pathogenesis of vitiligo.

Lastly, due to the highly admixed nature of the Brazilian population, we have corrected the results for population stratification, thus allowing us to detect potentially hidden associations, besides avoiding false-positive results.

## 2. Materials and Methods

### 2.1. Samples

Blood samples were collected from 410 unrelated volunteers from Ribeirão Preto, Southeastern Brazil, consisting of 367 healthy controls and 43 vitiligo cases, followed up at the Dermatology Outpatient Clinic of the University Hospital of Ribeirão Preto Medical School, University of São Paulo, between 2016 and 2018. No individual included among the healthy controls showed vitiligo signals. Moreover, vitiligo patients or control individuals with a history of autoimmune diseases were excluded from the study sample. Ethical approval was obtained from the Ethics Committee of Universidade de São Paulo (CAAE), #25696413.7.0000.5407, and all participants provided written informed consent.

### 2.2. Laboratory Analysis

DNA was extracted using a modified salting-out protocol [21]. NanoDrop^®^ ND-1000 (Thermo Fisher Scientific Inc., Waltham, MA, USA), agarose gel electrophoresis, and Qubit™ dsDNA BR Assay (Life Technologies, Carlsbad, CA, USA) were used for the evaluation of purity level, integrity, and concentration of the genomic DNA, respectively. Finally, all samples were normalized to 5 ng/µL to achieve an ideal sequencing library preparation concentration.

Sequencing libraries were prepared according to the manufacturer’s instructions using a customized Haloplex Target Enrichment System (Agilent Technologies, Inc., Santa Clara, CA, USA) protocol. A SureDesign tool (Agilent Technologies, Inc., Santa Clara, CA, USA) was employed to design a set of probes that ensured the capture of 488.658 bp, including the *HLA-G* (5′URR—upstream regulatory region, coding sequencing (CDS, only exons), and 3′UTR), and *LILRB1/2* (CDS, only exons) genes, as well as other loci of interest for the research group, such as the regions encompassing the SNP*for*ID 34-plex ancestry informative marker (AIM) set of SNPs [22].

DNA libraries were quantified using Qubit^®^ 2.0 Fluorometer (Thermo Fisher Scientific Inc., Waltham, MA, USA) and 2100 Bioanalyzer (Agilent Technologies, Inc., Santa Clara, CA, USA). Ultimately, a pool of DNA libraries of up to 96 samples was diluted to 16 pM and inserted as input for sequencing using the MiSeq Reagent Kit V3 (600 cycles) in the MiSeq Personal Sequencer (Illumina Inc., San Diego, CA, USA).

### 2.3. Bioinformatics Analysis

The *HLA-G*, *LILRB1*, and *LILRB2* mapping; genotype calling; and haplotyping strategies used here have already been published elsewhere [23,24]. Briefly, CutAdapt [25], hla-mapper version 2.2, function DNA, database version 2.1 [26], and GATK v.3.7 HaplotypeCaller in -ERC GVCF mode [27] were used for trimming adaptor sequences, alignment to the reference genome sequence (GRCh38/hg38) and genotype calling, respectively. Uncertainly, genotypes were interrogated using VCFx checkpl (www.castelli-lab.net/apps/vcfx (accessed on 1 October 2021)), with the minimum genotype likelihood set to 99.9% to retain only high-quality genotypes. GATK routine ReadBackedPhasing, using a minimal phase quality threshold of 500, coupled with a Bayesian probabilistic model implemented in the PHASE software [28], enabled taking some phase information from the paired-end reads, adding efforts in the identification of the most probable haplotypes. The phased VCF file was converted into *HLA-G* CDS sequences by using the hg38 reference sequence as a draft and replacing the correct nucleotide at each position, two sequences per sample, using the application vcfx (function fasta) (www.castelli-lab.net/apps/vcfx (accessed on 1 October 2021)). Using a local BLAST server with databases containing all known class I and II HLA CDS sequences described so far, downloaded from the IPD-IMGT/HLA database (https://www.ebi.ac.uk/ipd/imgt/hla/ (accessed on 1 October 2021)) version 3.31.0, the closest known *HLA-G* coding allele was defined for each haplotype.

The procedures applied for genotyping *LILRB1/2* exonic regions and the SNP*for*ID 34-plex AIMs were very similar to those used to *HLA-G*, except that BWA-MEM (Burrows–Wheeler) [29] instead of hla-mapper [26] was employed to the reference genome (GRCh38/hg38) alignment and that for SNP*for*ID 34-plex only genotype calling was performed. In addition, the procedures used for *HLA-G* haplotypic identification did not apply to these loci.

### 2.4. Statistical Methods

Based on the previously described approach [30], the SNP*for*ID 34-plex ancestry informative SNP panel was employed by STRUCTURE v.2.3.4 [31] to quantify the ancestral contributions of the present study samples. Afterward, the ancestry components obtained for each individual were included as covariates in a logistic regression model using PLINK v.1.9 [32] to correct the associations for population structure. Since the genetic risk depends on the type of inheritance, SNPs were analyzed using additive and dominant models.

The linkage disequilibrium (LD) pattern was evaluated by estimating the parameters *D’*, log of odds (LOD) scores, and *r^2^*. The haplotype blocks were defined by the confidence intervals method implemented in Haploview v.3.32 [33], excluding markers with a minor allele frequency (MAF) below 1% and a Hardy–Weinberg equilibrium (HWE) *p*-value lower than 0.05. Adherence of genotypic proportions to expectations under Hardy–Weinberg equilibrium, as well as the allelic frequencies, were evaluated using PLINK v.1.9 [32].

Single polymorphisms were first tested one by one for statistical association with the vitiligo risk using a traditional regression model. Due to the highly admixed nature of the Brazilian population, we adjusted the results for admixture proportions to avoid false-positive results arising from a population stratification bias [34] and to allow the detection of potentially hidden associations [35].

We then performed multivariate analyses considering several markers simultaneously using the machine learning method MDR [36] to screen for potential gene–gene interactions among the 213 variants from *HLA-G* and *LILRB1/2*.

Since the MDR efficiency can be increased by limiting the number of input features [37], the variants were LD-pruned before epistasis analysis to remove any multicollinearity between markers. PLINK v.1.9 [32] was used for pruning, considering the variance inflation factor (VIF) LD pruning routine that iteratively excludes individual variants that have a VIF > 2 with other variants (window size of 50 SNPs, shifting 5 SNPs at each step), using a minor allele frequency (MAF) threshold of 0.1. After pruning, 34 variants remained.

MDR reduces the dimensionality of multilocus information by testing all the possible combinations of multilocus genotypes and reports the ones exhibiting the best classification for disease risk [38]. Different approaches were tested: MDR v3.0.2 [39], generalized multifactor dimensionality reduction (GMDR) v.0.9 [40], and model-based multifactor dimensionality reduction (MB-MDR) [41].

The model selection and evaluation follow a similar strategy in the MDR v3.0.2 and GMDR v.0.9 methods. For developing a model, 9/10th of the data is employed, and a classification error is estimated at the end. Then, the remaining 1/10th of the data estimates the prediction error of the tested model. This procedure is repeated for each data piece, with classification and prediction errors obtained across all 10 runs. The ranking of the models relies on two parameters: the testing balanced accuracy (TBA), which indicates the degree of accuracy with which an interaction correctly classifies individuals as cases or controls, and the cross-validation consistency (CVC), which indicates the number of times an interacting set shows along the cross-validation subsets (Hastie et al. 2001). Single best models were selected from each of the two-marker and three-marker combinations. Among this set of best multifactor models, the combination of polymorphisms that maximizes both the TBA and the CVC was selected. Although very similar to the original MDR method, GMDR v.0.9 [40] classifies the individuals based on a residual-based score and provides the signal test indicating the significance of the identified model [40].

In contrast, some modifications are observed in the MB-MDR method [41]. First, the method does not split the data into training and learning sets. Model selection is based on the strength of association between a set of genotypes and the binary outcome of interest, using all the data, with the statistical significance of the models being assessed through a permutation testing strategy by randomizing the case/control status in the original dataset. In addition, risk categories are defined using a regression model, which allows adjustment for population stratification. Finally, it presents all significant genetic interactions and the best one. To ensure that the analyses were not influenced by chance or by initial conditions, each analysis was repeated 5 times using 5 different random seeds (permutation = 1.000; adjust = ancestry covariates; family = binomial; significance level = 0.05).

## 3. Results

The *HLA-G* variability was evaluated using massive parallel sequencing concerning the 5′URR extended distal region (−2635), CDS, and 3′UTR and the genetic diversity of *LILRB1*/*LILRB2* exonic regions in a sample of 410 individuals. A total of 110, 58, and 55 variation sites were identified in *HLA-G*, *LILRB1*, and *LILRB2*, respectively. After applying quality control criteria (genotyping call rate > 0.98 and Hardy–Weinberg test *p*-value ≥ 0.01), 103, 57, and 53 variants remained and were included in the association study with vitiligo risk.

The results of single-marker association analyses are presented in Table 1 (additive model) and Table 2 (dominant model). Interestingly, many of the associations detected before adjusting for population structure were no longer significant after taking ancestry composition into account in the regression model.

Among the SNPs significantly associated with the dominant model (Table 2), rs6932888 and rs6932596 are in strong LD with each other (*r^2^* ≥ 0.90).

Furthermore, multivariate analysis using different MDR-based methods was performed to identify possible gene–gene interactions affecting the disease risk. The three methods identified the same 2-SNP combination as the best model, providing the best prediction accuracy (Table 3, Figure 1).

Although 3-SNP combinations were also evaluated, their results were not shown since the 2-SNP combination results provided a better explanation of the vitiligo risk.

MB-MDR results also provided 20 different epistatic interactions significantly associated with the disease (Appendix A).

## 4. Discussion

Despite the latest advances in vitiligo, the role of the simultaneous genetic composition of *HLA*-G, *LILRB1*, and *LILRB2* in disease development remains to be elucidated. Once the inhibitory properties of HLA-G depend on the interaction with its receptors [18], the present study has evaluated both univariate and multivariate associations with vitiligo. The vitiligo pathogenesis has yet to be fully understood [42]. Several theories have been proposed to explain it, including the autoimmune, neural, biochemical, and genetic hypotheses. Although many of these factors may play essential roles in vitiligo pathogenesis, at least for nonsegmental vitiligo, the autoimmune hypothesis is currently the most accepted [6,43], as vitiligo often has autoimmune comorbidities and often responds to immunosuppressive treatments [44]. Furthermore, many of the 50 loci have already been associated with nonsegmental vitiligo showing immunomodulatory functions, while only a few genes have been linked to melanogenesis [45].

Among the associations already reported concerning this disease, those involving genes from the human leukocyte antigen (HLA) region, such as *HLA-A2*, *HLA-DR4*, and *HLA-DR7*, may be the leading cause of self-antigen recognition [45,46]. It has been suggested that HLA-G, a key immunoregulatory molecule, may play a role in vitiligo susceptibility [19,47,48]. This hypothesis is further supported by frequent *HLA-G* association with other autoimmune diseases, such as lupus erythematosus systemic [49,50], Crohn’s disease [51,52], multiple sclerosis [53,54], rheumatoid arthritis [55,56], and pemphigus vulgaris [57], along with HLA-G ectopic expression in skin pathologies [58,59].

HLA-G, an important nonclassical HLA-class Ib immunomodulatory molecule, plays its roles by binding to inhibitory receptors, such as leukocyte Ig-like receptors (LILRs, also called LIR, Ig-like transcript (ILT), or CD85): LILRB1 (LIR1/LILRB1/CD85j) and LILRB2 (LIR2/LILRB2/CD85d), along with KIR2DL4 (killer cell Ig-like receptor 2DL4) [60].

It has been shown that LILRB1 and LILRB2 are preferential ligands for HLA-G compared with classical MHCI proteins (HLA-A, HLA-B, HLA-C) once their affinity for the molecule is three- to fourfold higher [16]. Furthermore, polymorphisms in these loci may play a role in the LILR strength binding to MHC-I alleles [61]. However, no study has evaluated the role of *LILRB1* and *LILRB2* polymorphisms in developing vitiligo or the possible interactions between *HLA-G* and the genes encoding its receptors.

Accordingly, whether or not individual polymorphisms in *HLA-G*, *LILRB1*, and *LILRB2* were associated with susceptibility to vitiligo and whether or not there were some epistatic interactions between them were, therefore, investigated here. The association analysis of these polymorphisms with vitiligo gave rise to some SNPs associated with the disease, in either the additive (Table 1) or the dominant model (Table 2). Nevertheless, the studied population’s genetic structure significantly influenced this outcome once just a few SNPs remained associated after adjustment by ancestry. In contrast, others were only noticed after this adjustment was considered (rs272423), which may reflect an effect of the admixed nature of the Brazilian population [62], leading to false-positive associations with vitiligo. Such impact of population structure on the association results was already predicted in a previous study showing a strong influence of ancestry composition on the *HLA-G* haplotype distribution in a healthy Brazilian population sample [30].

Among the dominant model test results, three SNPs were associated with vitiligo in univariate analysis, even after adjustment by ancestry (Table 2). Two of them (rs6932888 and rs6932596) are in a complete LD one with each other (*D’* = 1, *r^2^* = 1), and the third one (rs9380142) is located in 3′UTR and also shows a strong allelic association with the first two SNPs (*r^2^* = 0.9 in both cases). They thus represent a single association signal. Considering the functional relevance of this 3′UTR variant, which has been previously associated with vitiligo [19] as well as several pathological conditions [63,64,65,66,67,68,69], and whose role in the *HLA-G* mRNA stability has been well established in vitro [63], we reinforce that this SNP is likely to be the causal variant directly associated with vitiligo susceptibility. This finding is consistent with the higher odds ratio and lower *p*-value observed for this SNP compared with the other two variants. The fact that this association was identified only under a dominant model agrees with a previous study suggesting that the rs9380142*G allele may behave dominantly [70]. Interestingly, the +3187G allele is observed only in the UTR-01 haplotype, which is associated with a higher production of HLA-G [71].

This study disclosed for the first time an association between rs272423 and vitiligo. Although rs272423 corresponds to a silent polymorphism with no apparent impact on the protein sequence, studies have shown evidence of the non-neutrality of synonymous codons in functional contexts, such as in the transcription factor (TF) binding regulation, leading to a change in gene expression levels [72,73]. The effect of this variant on TF binding was evaluated using HaploReg v4.1, which predicted that rs272423 alters the binding motif for the TFs ER alpha-a, LXR2, LXR3, NRSF, PLAG1, Rad21, and VDR3. Moreover, this SNP significantly changed the *LILRB1* gene expression in CEU-CHB-JPT lymphoblastoid cell lines, *p*-value = 9.6 × 10^−14^ [74].

Since HLA-G physically interacts with LILRB1 and LILRB2 to play its immunomodulatory role, the effect of one gene may only be perceived by considering the effects of the others [75]. Therefore, we conducted a study to survey the nucleotide variation of these three genes in the same set of individuals and explored the potential interactions of the polymorphisms identified and their effects on the risk of vitiligo using MDR-based methods specifically designed for the detection of epistasis [39].

The MDR method performs a data reduction strategy that classifies genotypes as high- or low-risk groups, reducing the predictors from *n* to one dimension [36]. All possible genotype combinations are tested, and the software reports the best model for disease risk classification or the model showing maximum TBA and CVC values [76]. The most significant model associated with vitiligo using MDR v.3.0.2 was a second-order combination between *HLA-G* rs9380142 and *LILRB1* rs2114511, accurately classifying 65% of tested individuals in 9 out of 10 validation intervals (Figure 1). According to GMDR v.0.9 [40], the same second-order combination (rs9380142/rs2114511) previously identified by MDR v.3.0.2 was selected as the best model for this disease (signal *p*-test = 0.0107), allowing for accurately classifying 64% of the individuals in all 10 validation intervals. Another extension to the MDR method is MB-MDR [41], which allows one to consider the ancestry composition in the association test. Instead of selecting the best model, it provides a set of combinations whose significance has been assessed by a permutation test. The same SNP combination was again identified as the best model for the disease (permutation *p*-value = 0.0130).

The first SNP rs9380142 in the epistatic interaction identified by all methods is located at *HLA-G* 3′UTR, 4-bp upstream to the AUUUA pentameter, a motif related to mRNA degradation. When the rs9380142+3187A allele is present, a shorter *HLA-G* mRNA half-life in vitro is observed, possibly due to an adenine increase in this AU-rich motif [63]. Although not as extensively investigated as the 14 bp insertion/deletion (INDEL) and +3142, +3187 was already evaluated for preeclampsia [63,67], septic shock [77], malaria [65], leprosy [66], celiac disease [68], rheumatoid arthritis [70], schizophrenia [78], HTLV1 infection [69], bipolar disorder [79], epithelial ovarian cancer [80], and vitiligo [19]; it is observed in the latter similar association pattern, which is expected once there is a sample overlap between both studies.

The second SNP rs2114511 involved in the epistatic interaction corresponds to a synonymous variant from the *LILRB1* gene. Although no association involving this SNP has been reported so far, according to HaploReg v.4.1, this site resembles a TF binding motif activated in the T-lymphocyte subtypes, monocytes, B, NK, spleen, and GM12878 lymphoblastoid cells. The SNP rs2114511 is an expression quantitative trait locus (eQTL) that regulates the LILRB1 expression [81] and that alters the AP2, CTCF, Roas2, and Zic4 binding motifs. Except for Roas2, they are all expressed in the skin (Human Protein Atlas portal—http://www.proteinatlas.org (accessed on 1 December 2021)). The GTEx portal (https://gtexportal.org/home/ (accessed on 1 December 2021)) indicates that the variant rs2114511/C is associated with higher expression of LILRB1 in the lung. Lastly, rs2114511 is a binding site for EBF transcription factor 1 (EBF1) in the B-cell-derived cell line (GM12878 lymphoblastoid cells; ENCODE Project Consortium, 2011).

EBF1 is essential for the development of B lymphocytes once it affects the activating factor receptor (BAFF-R) and B-cell receptor (BCR)-dependent Akt pathways [82]. It has been shown that variation in signaling intensity through the B-cell receptor (BCR), CD40R, and BAFF could lead to immunological dysregulation and self-tolerance loss, besides modulating the B cells’ destiny [83]. B-cell-activating factor (BAFF) is a tumor necrosis factor (TNF) ligand family member. It stimulates the interleukin (IL-2) and interferon (INF)-γ production in the CD4+ T cells and the peripheral blood mononuclear cell proliferation [84]. In agreement with the latter data, immunoglobulin G (IgG) and immunoglobulin M (IgM) against melanocytes were found in 80% of vitiligo patients [85].

Despite the genetic variant usually associated with higher levels of HLA-G (rs9380142/G) being present in the interaction, the second variant (rs2114511/G) is associated with lower expression of LILRB1 [86]. In other words, it is no use to have the molecule (HLA-G) if we do not have the receptor (LILRB1) once that immune response is reached from binding between both molecules. These findings agree with a study that has observed a negative correlation between HLA-G expression and vitiligo risk [87] and with the general idea that a lower HLA-G expression or function would be correlated with autoimmunity.

Considering that de novo HLA-G expression at high levels could support an immunosuppressive response in pathological conditions, such as cancer and viral infections, and the HLA-G downregulation expression could impair the HLA-G mediated control of the immune response leading to autoimmunity [88], the correlation between higher HLA-G and lower LILRB1 levels and vitiligo is here suggested as a hypothesis of a mechanism associated with impaired control of immune response at the skin level.

## 5. Conclusions

It may be concluded that two polymorphisms in *HLA-G* (rs9380142) and *LILRB1* (rs272423) are independently associated with nonsegmental vitiligo in the Brazilian population after adjustment for ancestry. Moreover, a consistent second-order combination involving *HLA-G* (rs9380142) and *LILRB1* (rs2114511) was identified, indicating an epistatic interaction role between *HLA-G* and *LILRB1* alleles in vitiligo pathogenesis. Nonetheless, the mechanism by which this combination affects the autoimmune response development remains to be evaluated in further studies. Finally, some limitations of the study must be indicated, such as the small vitiligo sample size, the larger proportion of females in the vitiligo group, and the lack of studies supporting a direct link between our findings and the disease development.

## Figures and Tables

**Figure 1 cells-12-00630-f001:**
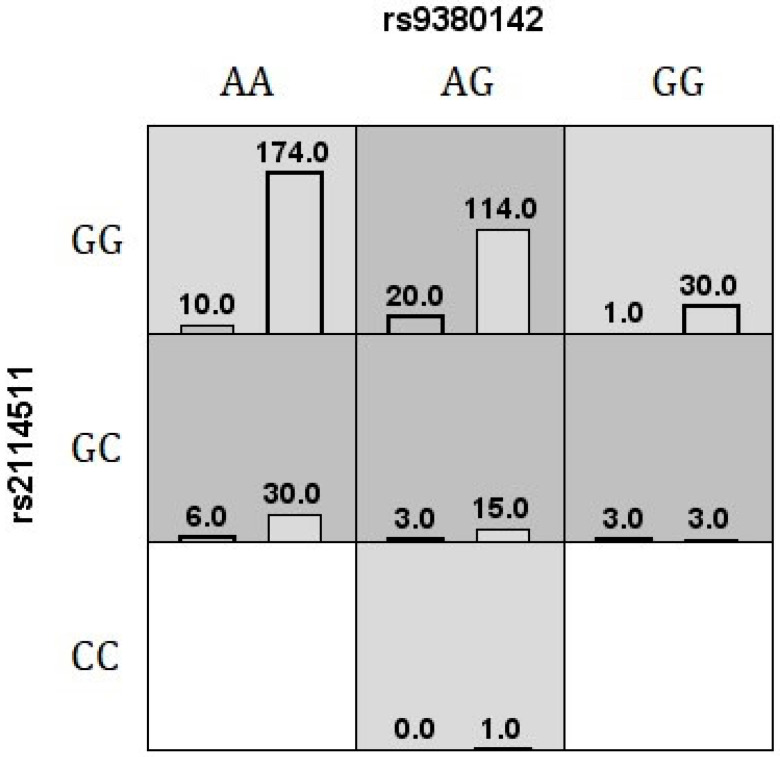
The best MDR v.3.0.2 interaction model for vitiligo risk is shown. The number of cases (left bars) and controls (right bars) are illustrated for each genotype combination. The white cells are labeled as unknown, light gray cells are labeled as low risk, and dark gray cells are labeled as high risk.

**Table 1 cells-12-00630-t001:** Significant results of single-marker association tests of *HLA-G*, *LILRB1*, and *LILRB2* SNPs with vitiligo in the Brazilian population under an additive model, before and after adjustment for ancestry.

Additive Model			Before Adjustment		After Adjustment	
GENE	SNP	Allele	OR (L95–U95)	*p*	OR (L95–U95)	*p*
*HLA-G*	**rs17875403 ^1^**	T	8.902 (1.221–64.890)	0.0309	-	-
*LILRB2*	**rs10405713 ^1^**	C	2.357 (1.091–5.094)	0.0292	-	-
*LILRB2*	**rs373032 ^1^**	T	0.485 (0.251–0.938)	0.0315	-	-
*LILRB1*	rs10427127	C	1.957 (1.082–3.538)	0.0263	-	-
*LILRB1*	**rs10425827 ^1^**	G	2.210 (1.160–4.209)	0.0159	-	-
*LILRB1*	rs61739173	A	2.173 (1.088–4.337)	0.0278	-	-
*LILRB1*	**rs2114511 ^1^**	C	2.349 (1.156–4.775)	0.0182	-	-
*LILRB1*	**rs272423 ^1^**	C	-	-	0.5611 (0.331–0.950)	0.0316

^1^ SNPs highlighted in bold are associated under both additive and dominant models. OR = odds ratio. CI = confidence interval. *p*-Value ≤ 0.05 was considered significant.

**Table 2 cells-12-00630-t002:** Significant results of single-marker association tests of *HLA-G*, *LILRB1*, and *LILRB2* SNPs with vitiligo in the Brazilian population under a dominant model, before and after adjustment for ancestry.

Dominant Model			Before Adjustment		After Adjustment	
GENE	SNP	Allele	OR (L95–U95)	*p*	OR (L95–U95)	*p*
*HLA-G*	rs6932888	C	1.914 (1.004–3.649)	0.0485	2.022 (1.052–3.888)	0.0348
*HLA-G*	rs6932596	T	1.914 (1.004–3.649)	0.0485	2.022 (1.052–3.888)	0.0348
*HLA-G*	**rs17875403 ^1^**	T	8.902 (1.221–64.890)	0.0301	-	-
*HLA-G*	rs9380142	G	2.112 (1.101–4.053)	0.0246	2.225 (1.149–4.308)	0.0176
*LILRB2*	**rs10405713 ^1^**	C	2.660 (1.075–6.580)	0.0343	-	-
*LILRB2*	**rs373032 ^1^**	T	0.434 (0.208–0.906)	0.0263	-	-
*LILRB1*	**rs10425827 ^1^**	G	2.130 (1.010–4.492)	0.0470	-	-
*LILRB1*	**rs2114511 ^1^**	C	2.512 (1.209–5.219)	0.0135	-	-
*LILRB1*	**rs272423 ^1^**	C	-	-	0.502 (0.261–0.965)	0.0388

^1^ SNPs highlighted in bold are associated under both additive and dominant models. OR = odds ratio. CI = confidence interval. *p*-Value ≤ 0.05 was considered significant.

**Table 3 cells-12-00630-t003:** Significant second-order interactions simultaneously identified by all MDR methods applied.

MDR Method	Best 2 Order Model	TBA, CVC	*p*-Value
MDR v3.0.2	rs9380142, rs2114511	0.65, 9/10	-
GMDR v.0.9	rs9380142, rs2114511	0.64, 10/10	0.0107 ^a^
MB-MDR (before adjustment)	rs373032, rs2114511	-	0.0020 ^b^
MB-MDR (after adjustment)	rs9380142, rs2114511	-	0.0130 ^b^

^a^ signal *p*-test. ^b^ permutation *p*-value.

## Data Availability

The genetic data are available in Figshare online repository upon submission codes 14959416 (LILRB1) and4959413 (LILRB2).

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
