# Peer review of "Evidence for Epistatic Interaction between HLA-G and LILRB1 in the Pathogenesis of Nonsegmental Vitiligo"

_cells, 2023, doi:10.3390/cells12040630_

Round 1

Reviewer 1 Report

The manuscript describes an interesting study evaluating the genetic association of SNPs in HLA-G and its receptors, LILRB1, and LILRB2, with non-segmental vitiligo. A variety of statistical methods, including logistic regression along with adjustment by ancestry and the Multifactor Dimensionality Reduction (MDR) approach, were adopted to investigate the association of these SNPs individually and in combination with vitiligo. Generally, their methods are sound, and their statements are convincing. The knowledge gained here will benefit our audience. However, I did have a question listed below for the authors to address before it can be published in our journal.

Given that the SNP rs9380142 is expected to reduce the half-life of the HLA-G mRNA through its potential impact on mRNA degradation, it is likely the SNP will decrease the protein expression level of HLA-G. If this is the case, why the authors suggested the correlation between higher HLA-G and LILRB1 levels with vitiligo?

Author Response

It was hypothesized that 3’UTR haplotypes containing the +3187 G variants might be associated with high expression of HLA-G (Donadi et al. 2011). Furthermore, the allele G is observed only in the UTR-1 haplotype (Castelli et al. 2014), which has been associated with a higher production of HLA-G (Martelli-Palomino et al. 2013; Poras et al. 2017). However, we should evaluate the genetic variants' effects together once the observed outcomes could be divergent.

Our explanation for this effect is related to the presence of the *GG alleles in the second site (rs2114511). Although no extensive studies have explored the effects on the protein expression of LILRB1 genetic variants, the GTEx portal (https://gtexportal.org/home/) indicates that variant rs2114511/C is associated with higher expression of LILRB1 in the lung (association also observed by Castelli et al. 2021). Therefore, despite the genetic variant usually associated with higher levels of HLA-G (rs9380142/G) being present in the interaction, the second variant (rs2114511/G) is associated with lower expression of LILRB1. In other words, it is no use to have the molecule (HLA-G) if we do not have the receptor (LILRB1) once that immune response is reached from binding between both molecules. This issue was addressed in lines 358-360.

Reviewer 2 Report

Thanks for the great efforts in completion of the clinical study and writing this manuscript.

-The manuscript is written in a sophisticated way with too much unnecessary details, it would be better to simplify and concise the data presentation as much as possible.
- The sample size included a low number of vitiligo patients in relation to apparently healthy volunteers. Furthermore, the authors didn’t clarify the family history of vitiligo among these healthy volunteers. 

Author Response

Thank you for the comment. We have revised the manuscript to eliminate unnecessary details, such as in the Results section.

We could compute the adequate sample size to achieve 80% statistical power using statistical formulae or a web browser program, such as Genetic Power Calculator developed by Purcell et al. (https://zzz.bwh.harvard.edu/gpc/), for case-control studies. However, empirical population genomic studies have yet to be able to define optimal sampling strategies. Moreover, our sampling approach included all vitiligo patients followed up in a large time window (from 2016 to 2018) at the Dermatology Outpatient Clinic of the University Hospital of Ribeirão Preto Medical School, University of São Paulo that agreed to take part in our study.  

Therefore, we are aware that this is a weakness of the study. However, considering that if an association is yet significant even when a small sample size is employed, we can trust that it is an actual result. Still, at the same time, it is possible that our study fails to identify other genuine associations that cannot be noticed due to the small sample size. One strategy to statistically compensate for this low number of vitiligo patients is to raise the number of control subjects studied, which was adopted in our study.

Finally, we want to clarify that no individual included among the healthy volunteers showed vitiligo signals. Moreover, any subject showing an autoimmune family history (either from case or control groups) was excluded from our study. We included a statement in the Materials and Methods section clarifying this point (lines 86-88).

Author Response

Major Concerns:

I completely agree with your concerns that, at this point, it is hard to tell how they are involved in vitiligo.

Considered an immune checkpoint molecule, HLA-G interacts with the specific inhibitory receptors Leukocyte Immunoglobulin (Ig)-like Receptors (LILR) LILRB1 and LILRB2, leading to down-regulation of innate and adaptive immunity. Due to the critical function of HLA-G-LILRB1 recognition for the modulation of immune tolerance, the specific combinations of alleles in these genes could provoke a greater or lesser inhibition of NK cells, antigen-presenting cells, T and B lymphocytes and, consequently, of the immune response. Although these findings do not provide a direct relationship that helps explain the etiopathogenesis of the disease, identifying causal genetic variants may provide the basis for future elucidation of these immunopathogenic mechanisms (Spritz and Andersen 2017; Boniface et al. 2018). This issue was addressed in lines 380-381.

Minor Concerns:

Thank you for the observation. We have checked again for grammatical errors in the text.

Reviewer 4 Report

Dear authors,

This research article is generally well-written, but it needs major editings. 

Editing suggestions;

1) Abstract section should be more detailed.

2) English grammar editing required.

3) It is recommended that the Introduction, material and methods, results, discussion sections should be in bold in the form of a heading at the beginning of the paragraphs.

4) Female dominance in the study group is very high, Female dominance should be stated as limitation.

5) In the Results section; do not repeat in the text information that is already present in the tables.

6) In the discussion part, comments should not be given without discussing the results with the previous study data in the literature.

Best wishes...

Author Response

1) Abstract section should be more detailed.

Thank you for the comment. We completely agree with that, but we must follow the journal's instructions. According to the instructions, the abstract section should be limited to 200 words.

2) English grammar editing is required.

Thank you for the observation. We have checked again for grammatical errors in the text.

3) It is recommended that the Introduction, material and methods, results, and discussion sections should be in bold in the form of a heading at the beginning of the paragraphs.

Thank you for the observation. We have made these modifications in the text.

4) Female dominance in the study group is very high, Female dominance should be stated as a limitation.

We recognize that the number of case samples is a limitation of our study, so the results should be carefully observed. Moreover, the dominance of females among patients is also noteworthy. This is certainly because vitiligo is a skin condition that affects women's quality of life more significantly than men's since it involves feelings about appearance and self-esteem (Amer & Gao, 2016). Unfortunately, our sample size does not allow for identifying any influence of gender on associations involving HLA-G, LILRB1, and LILRB2. This issue was addressed in lines 379-380.

5) In the Results section, do not repeat in the text information that is already present in the tables.

Thank you for the observation. We have made modifications in the text that address this issue.

6) In the discussion part, comments should not be given without discussing the results with the previous study data in the literature.

Best wishes...

Thank you for the observation.  We have addressed this issue in the Discussion section.

Round 2

Reviewer 3 Report

The authors have tried to address the comments as required. The manuscript can now be accepted for its publication.